

# Application of a MALDI-TOF analysis platform (ClinProTools) for rapid and preliminary report of MRSA sequence types in Taiwan

Hsin-Yao Wang[1,2,*], Frank Lien[1,*], Tsui-Ping Liu[1], Chun-Hsien Chen[3], Chao-Jung Chen[4,5] and Jang-Jih Lu[1,6,7]

[1] Department of Laboratory Medicine, Chang Gung Memorial Hospital at Linkou, Tauyuan, Taiwan
[2] Ph.D. Program in Biomedical Engineering, Chang Gung University, Taoyuan, Taiwan
[3] Department of Information Management, Chang Gung University, Taoyuan, Taiwan
[4] Graduate Institute of Integrated Medicine, China Medical University, Taichung, Taiwan
[5] Proteomics Core Laboratory, China Medical University Hospital, Taichung, Taiwan
[6] Department of Medical Biotechnology and Laboratory Science, Chang Gung University, Taoyuan, Taiwan
[7] School of Medicine, Chang Gung University, Taoyuan, Taiwan
* These authors contributed equally to this work.

Corresponding authors
Chao-Jung Chen,
cjchen@mail.cmu.edu.tw
Jang-Jih Lu, jjlpcp@cgmh.org.tw

## ABSTRACT

**Background:** The accurate and rapid preliminarily identification of the types of methicillin-resistant *Staphylococcus aureus* (MRSA) is crucial for infection control. Currently, however, expensive, time-consuming, and labor-intensive methods are used for MRSA typing. By contrast, matrix-assisted laser desorption ionization time-of-flight mass spectrometry (MALDI-TOF MS) is a potential tool for preliminary lineage typing. The approach has not been standardized, and its performance has not been analyzed in some regions with geographic barriers (e.g., Taiwan Island).
**Methods:** The mass spectra of 306 MRSA isolates were obtained from multiple reference hospitals in Taiwan. The multilocus sequence types (MLST) of the isolates were determined. The spectra were analyzed for the selection of characteristic peaks by using the ClinProTools software. Furthermore, various machine learning (ML) algorithms were used to generate binary and multiclass models for classifying the major MLST types (ST5, ST59, and ST239) of MRSA.
**Results:** A total of 10 peaks with the highest discriminatory power ($m/z$ range: 2,082–6,594) were identified and evaluated. All the single peaks revealed significant discriminatory power during MLST typing. Moreover, the binary and multiclass ML models achieved sufficient accuracy (82.80–94.40% for binary models and >81.00% for multiclass models) in classifying the major MLST types.
**Conclusions:** A combination of MALDI-TOF MS analysis and ML models is a potentially accurate, objective, and efficient tool for infection control and outbreak investigation.

# INTRODUCTION

Since their emergence in the 1960s, methicillin-resistant *Staphylococcus aureus* (MRSA) infections have been a major health care concern worldwide (*Chen & Huang, 2014*; *Walter et al., 2015*; *Wang et al., 2010*). Many epidemiological studies have revealed that different multilocus sequence types (MLST) present specific characteristics such as virulence gene profiles (*Recker et al., 2017*; *Schuenck et al., 2012*; *Wang et al., 2009*, *2010*, *2012*). Understanding the evolution of MRSA lineages and the origin of infection is crucial in outbreak investigation. Molecular typing methods, such as pulsed-field gel electrophoresis and MLST, are highly expensive and labor intensive for epidemiological studies. Hence, the application of these methods in clinical practice is limited (*Struelens et al., 2009*). Sequence based typing method has been widely used since the past decade, and it provides adequately high resolution for confirming transmission. However, in regions with few medical resources and financial constraints, it remains relatively impracticable (*Harris et al., 2013*; *Koser et al., 2012*; *Schwarze et al., 2018*).

Recently, matrix-assisted laser desorption ionization time-of-flight mass spectrometry (MALDI-TOF MS) has been used in many clinical microbiology laboratories. This method can be used to identify bacterial species effectively and rapidly (*Ge et al., 2016*). The peptide or protein MS fingerprint of each bacterium can be generated and stored in a bacterial library for species identification. In addition, MALDI-TOF MS also provides an alternative solution for molecular typing methods (e.g., MLST) (*Lartigue, 2013*; *Lu et al., 2012*) and has the potential to offer lineage typing up to the subspecies level.

Studies that have adopted the MALDI-TOF MS approach (*Lasch et al., 2014*; *Sauget et al., 2017*; *Ueda et al., 2015*) have reported varying results; it may be due to several reasons. First, the predominant MRSA lineages in different areas are different and the discriminatory power of MALDI-TOF might therefore differ when compared between different MRSA lineages. Second, the bacterial MS fingerprints of isolates from one area may not match those of isolates from other areas. Third, in many of the published works, the MALDI-TOF mass spectra have been assessed manually. An objective, standardized, and automated protocol has not been widely applied thus far (*Camoez et al., 2016*). Finally, the data obtained from MALDI-TOF mass spectra are relatively complicated and may be analyzed by a wide variety of bioinformatics tools. A manual approach cannot ensure a consistently high-quality output, because of the potential for large interindividual or intraindividual variation in the interpretation of the data. Therefore, a reliable analysis platform is necessary to ensure comparability between reports in clinical practice. Although Staphylococcus protein A typing provides comparable performance as MLST typing with less cost and time, our study has adopted MLST typing because we aim to demonstrate the possibility to implement our method beyond *S. aureus* (*Crisostomo et al., 2001*; *O'Hara et al., 2016*). To validate the use of MALDI-TOF mass spectra in classifying MLST types of MRSA in Taiwan, we used the ClinProTools software for analyzing MALDI-TOF mass spectra to generate classification models of MRSA lineages. Accordingly, clinical microbiology laboratories may rapidly provide preliminary typing reports of MRSA, which may be further confirmed using a

sequence-based method, thus enabling clinical practitioners to exclude an outbreak or transmission in time.

## MATERIALS AND METHODS

### Bacterial lineages

This study included 306 convenience non-duplicate MRSA lineages isolated from multiple reference hospitals in Taiwan, mainly through the Surveillance of Multicenter Antimicrobial Resistance in Taiwan (SMART) program (*Wang et al., 2012*). The SMART program consecutively collected MRSA isolates from ten medical centers throughout Taiwan from March to August 2003 (*Ho et al., 2010*). All the lineages in this study were convenience samples which had been recovered from blood cultures. All duplicate isolates were removed from the study. After procuring the cultures from bacterial banks, tests were performed again to confirm the characteristics of each lineage. The identification of *S. aureus* was based on colony morphology, microscopic examination, a coagulase test, a catalase test, and MALDI-TOF mass spectra.

### Bacterial identification through MALDI-TOF MS

The fresh bacterial colonies that were grown on blood agar plates for 24 h were picked up and smeared onto a MALDI steel target plate, forming a thin film of colonies. Next, one μL of 70% formic acid was introduced on the film and dried at room temperature. Subsequently, one μL of the matrix solution (i.e., 50% acetonitrile containing 1% α-cyano-4-hydroxycinnamic acid and 2.5% trifluoroacetic acid) was introduced on the film again. The sample-matrix was dried at room temperature before analyzing it through MS for data acquisition. Mass spectrum analysis was performed using a MicroFlex LT mass spectrometer (Bruker Daltonik GmbH, Bremen, Germany) with linear positive model, and the analytic region was 2,000–20,000 Da. For each sample, 240 laser shots (at frequency of 20 Hz) were collected, and a Bruker Daltonics Bacterial test standard (Bruker Daltonik GmbH, Bremen, Germany) was used for calibration and as the control with the linear positive model. The procedures were conducted according to the manufacturer's instructions, which have been detailed in previous studies (*Ge et al., 2016*; *Lu et al., 2012*). The results of mass spectrum from the MALDI Biotyper 3.1 software (Bruker Daltonik GmbH, Bremen, Germany) were compared with those in the database and assigned scores. Peaks with scores >2 were further selected for peak signal analysis. The lineages were randomly divided into batches, and the analyses were conducted on different days to avoid a possible batch effect.

### Multilocus sequence types

We sequenced the lineages for seven housekeeping genes, namely carbamate kinase (*arcC*), shikimate dehydrogenase (*aroE*), glycerol kinase (*glpF*), guanylate kinase (*gmk*), phosphate acetyltransferase (*pta*), triosephosphateisomerase (*tpi*), and acetyl coenzyme A acetyltransferase (*yqiL*). The sequencing results of these genes were compared with those in the *S. aureus* MLST database (http://saureus.mlst.net/) to acquire an allelic number and a sequence type (*Enright et al., 2000*).

## MALDI-TOF MS spectra analysis

MALDI-TOF mass spectra of the MRSA lineages were fed into the ClinProTools™ software (version 3.0, Bruker Daltonik GmbH, Bremen, Germany) in batches. The data preprocessing steps, including baseline subtraction, smoothing, and recalibration, were set as default for all analyses (*Bruker Daltonik GmbH, 2011*; *Camoez et al., 2016*; *Zhang et al., 2015*). ClinProTools is a widely used software developed by Bruker (Bruker Daltonik GmbH, Bremen, Germany). It has been used in MALDI-TOF data analysis for MRSA lineages typing (*Camoez et al., 2016*; *Zhang et al., 2015*). Characteristic peaks among various MLST types were selected and sorted through several statistical tests, including the *t*-test, analysis of variance (ANOVA), the Wilcoxon or Kruskal–Wallis (W/KW) test, and the Anderson–Darling (AD) test. A *P*-value of 0.05 was set as the cutoff. If *P* was <0.05 in the AD test, a characteristic peak was selected if the corresponding value of *P* in the W/KW test was also <0.05. When *P* was ≥0.05 in the AD test, then a characteristic peak was selected if the corresponding value of *P* in ANOVA was also <0.05 (*Stephens, 1974*).

## Generation and validation of classification models

Classification models of the major MLST types (ST5, ST59, and ST239) were generated using the machine learning (ML) algorithms in ClinProTools, namely QuickClassifier (QC), Supervised Neural Network (SNN), and Genetic Algorithm-K Nearest Neighbor (GA-KNN). The description and setting of the ML models are detailed in the ClinProTools user manual (*Bruker Daltonik GmbH, 2011*). All the peaks in the spectra were used in model generation. The W/KW test was used to sort peaks during selection. For GA-KNN, GA was used as a method for supporting feature selection, where the maximum number of best peaks was set as 30, and the maximum number of generations was set as 50. The numbers of the nearest neighbors evaluated in the GA-KNN algorithm were 1, 3, and 5–7 for each binary classification. To avoid over fitting, we used fivefold cross validation to obtain an unbiased statistical measurement of performance. Accordingly, the data were split into five subsets in a randomized manner. Each subset would serve as the validation set for the model trained by the remaining four subsets iteratively. Classification accuracy was obtained from the average of the five evaluations. Consequently, the bias of over fitting could be avoided using fivefold cross validation.

## Statistical analysis

The AD test is used to test for a normal distribution of peak intensity in ClinProTools.

In the AD test when $P \leq 0.05$ (i.e., the data distribution did not follow normal distribution), the W/KW test was used as the statistical method to select discriminative peaks. In the W/KW test, if $P \leq 0.05$, a rank-based multiple test procedure was used for post hoc analysis to conduct paired comparisons in the non-normal distributed data and calculate the simultaneous confidence intervals with Tukey-type contrasts (*Konietschke, Hothorn & Brunner, 2012*). For evaluating the performance of various ML models, accuracy, sensitivity, and specificity were used as the metrics.

**Table 1 Characteristic MALDI-TOF MS peaks in different MLST types of MRSA.**

| Mass | DAve | PW/KW | PAD | Average of peak intensity | | | |
|---|---|---|---|---|---|---|---|
| | | | | ST5 | ST59 | ST239 | Other ST |
| 2082.13 | 4.28 | <0.000001 | <0.000001 | 6.13 | 3.84 | 1.85 | 3.24 |
| 2415.79 | 38.19 | <0.000001 | <0.000001 | 24.57 | 6.13 | 44.32 | 19.72 |
| 2430.49 | 18.13 | <0.000001 | <0.000001 | 20.12 | 3.86 | 21.99 | 6.23 |
| 2880.03 | 11.2 | <0.000001 | <0.000001 | 1.15 | 1.88 | 12.35 | 5.38 |
| 2980.45 | 6.85 | <0.000001 | <0.000001 | 7.4 | 2.47 | 9.32 | 7.88 |
| 3276.8 | 2.94 | <0.000001 | <0.000001 | 2.61 | 3.73 | 0.79 | 2.08 |
| 3876.96 | 6.24 | <0.000001 | <0.000001 | 0.75 | 5.5 | 6.33 | 7 |
| 3892.91 | 4.91 | <0.000001 | <0.000001 | 5.61 | 0.69 | 0.87 | 2.77 |
| 6553.03 | 6.23 | <0.000001 | <0.000001 | 6.28 | 9.32 | 3.1 | 6.15 |
| 6593.54 | 4.33 | <0.000001 | <0.000001 | 1.98 | 3 | 6.3 | 5.48 |

Notes:

DAve, difference between the maximal and the minimal average peak intensity of all classes; PW/KW, $P$-value obtained through Wilcoxon/Kruskal–Wallis test; PAD, $P$-value obtained through Anderson–Darling test; MALDI-TOF MS, matrix-assisted laser desorption ionization time-of-flight mass spectrometry.

$$\text{Accuracy} = \frac{TP + TN}{TP + FP + TN + FN}$$

$$\text{Sensitivity} = \frac{TP}{TP + FN}$$

$$\text{Specificity} = \frac{TN}{TN + FP}$$

where TP, TN, FP, and FN represent the number of true positives, true negatives, false positives, and false negatives, respectively.

## RESULTS

Six MLST types of MRSA were identified in the isolates, namely ST5 ($n = 40$), ST45 ($n = 8$), ST59 ($n = 62$), ST239 ($n = 179$), ST241 ($n = 12$), and ST573 ($n = 5$). The isolates ST5, ST59, and ST239 were considered the major MLST types of MRSA because they exhibited numerous lineages. The isolates ST45, ST241, and ST573 were categorized together as other ST types.

### Characteristic peaks for discrimination among various MLST types

The characteristic peaks were sorted using the corresponding $P$-values obtained in the (W/KW) test because the AD test revealed $P < 0.05$ (Table 1). The top 10 characteristic peaks, sorted using the W/KW test results, were selected for further statistical evaluation. The 10 peaks ranged from $m/z$ 2,082 to 6,594 (Table 1). All the 10 selected peaks revealed $P < 0.000001$ in the W/KW test, thus indicating that they were informative and discriminative peaks in the MLST type classification.

In Table 1, the distribution of these peaks over various MLST types was further evaluated. High expression levels of $m/z$ 2,430 and 3,893 and a low expression level of $m/z$ 3,877 indicated the fingerprint of ST5; low expression levels of $m/z$ 2,416, 2,980,

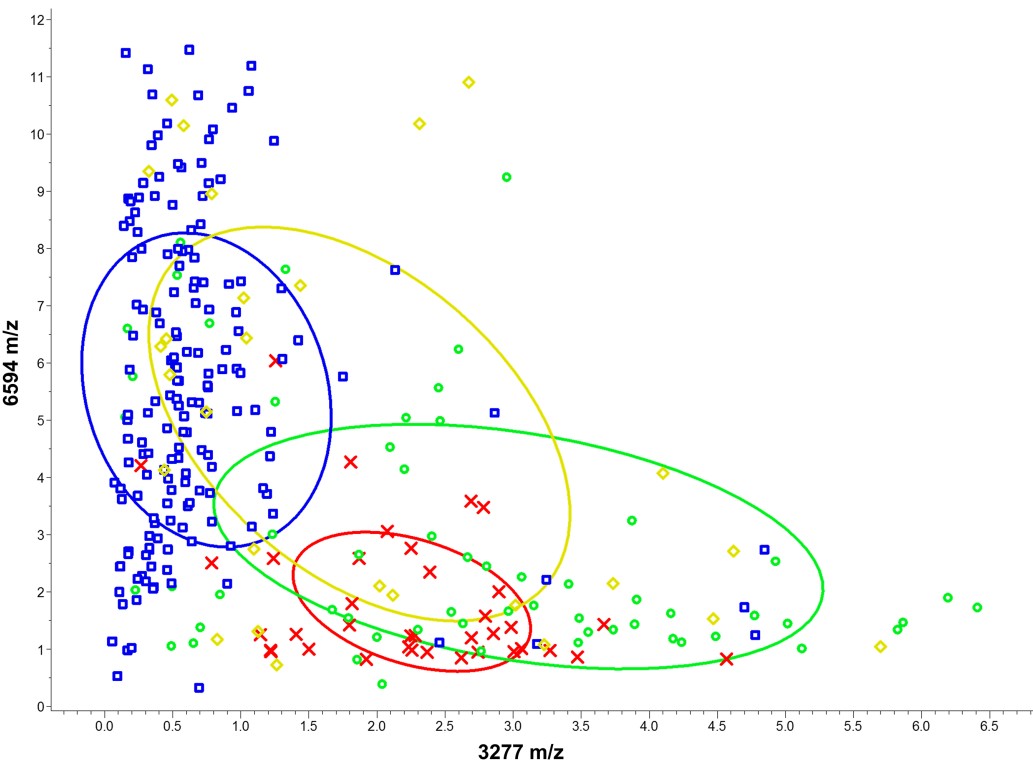

**Figure 1  Scatter plot of various MLST types isolates.** The peaks at *m/z* 3,277 and *m/z* 6,594 served as the *x*- and *y*-axes, respectively. The intensities of the characteristic peaks were expressed in arbitrary intensity units. The ellipses represent the 95% confidence intervals of peak intensities for each MLST type. In the preliminary analysis, the various MLST types could not be satisfactorily separated based only on two characteristic peaks. ST5: red crosses, ST59: green circles, ST239: blue squares, and other ST types: yellow diamonds.

and 3,893 indicated the fingerprint of ST59; high expression levels of *m/z* 2,416 and 2,880 and low expression levels of *m/z* 3,277 and 3,893 indicated the fingerprint of ST239. Based on 10 informative peaks, the odds ratio of different ST-pairs was calculated to determine the association between the peaks and respective ST-pairs (Table S1). Furthermore, the distribution of the isolates was plotted according to peak intensity of *m/z* 3,277 (*x*-axis) and *m/z* 6,594 (*y*-axis) (Fig. 1). The peaks at *m/z* 3,277 and 6,594 were the top two characteristic peaks among the ST types (Fig. 1). The scatter plot figures may be seen as a projection of a high-dimensional scatter space with various dimensions of the *m/z* peaks.

The average intensities of the 10 peaks of these lineages are further illustrated in Fig. 2. These results demonstrate the specific characteristics and patterns of peaks in the different MLST types. Specifically, the distribution of the ST59 lineages and ST239 lineages showed satisfactory separation (Fig. 1); ST5 lineages could also be distinguished from ST239 lineages based on distribution (Fig. 1). By contrast, ST5 lineages and ST59 lineages could not be satisfactorily discriminated using information from the peaks *m/z* 3,277 and 6,594 only (Fig. 1). Similarly, the lineages of other minor ST types mixed with other major ST types (i.e., ST5, ST59, and ST239) on the scatter plot (Fig. 1).
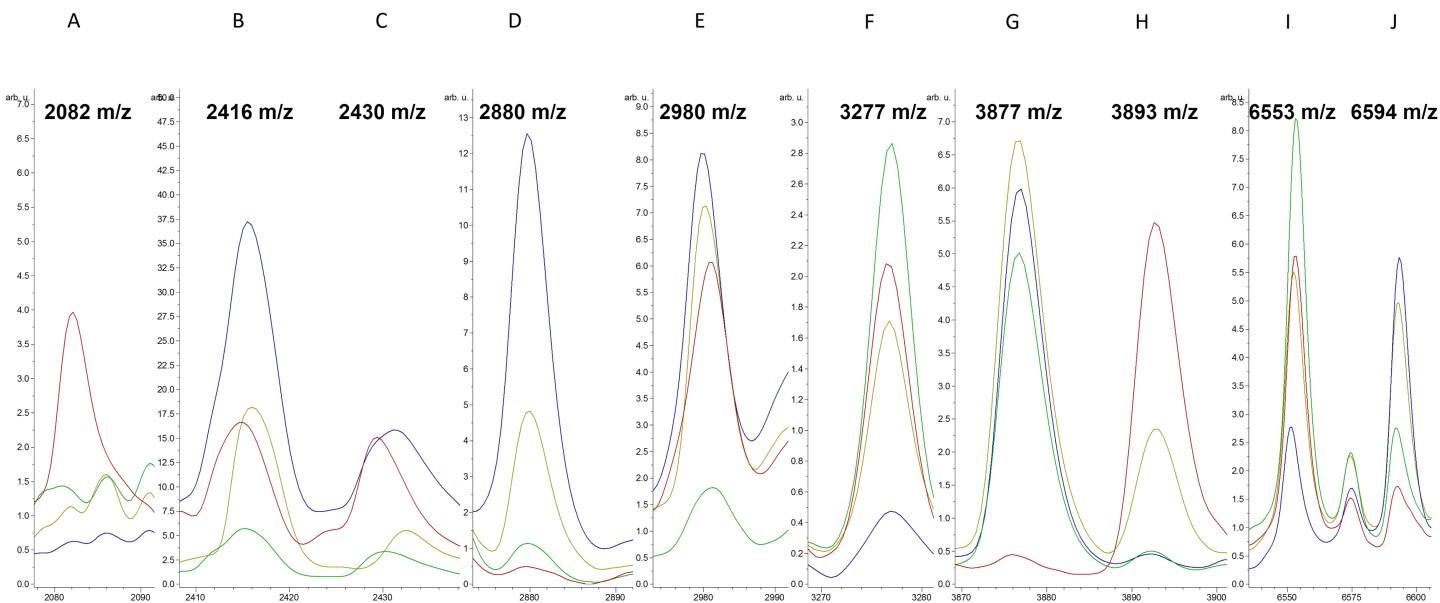

**Figure 2 Average spectra of characteristic peaks among various MLST types.** Intensities of characteristic peaks (*m/z* 2,082, 2,416, 2,430, 2,880, 2,980, 3,277, 3,877, 3,893, 6,553, and 6,594, from A to J, respectively) in ST5 (red), ST59 (green), ST239 (blue), and other ST types (yellow) expressed in arbitrary intensity units.

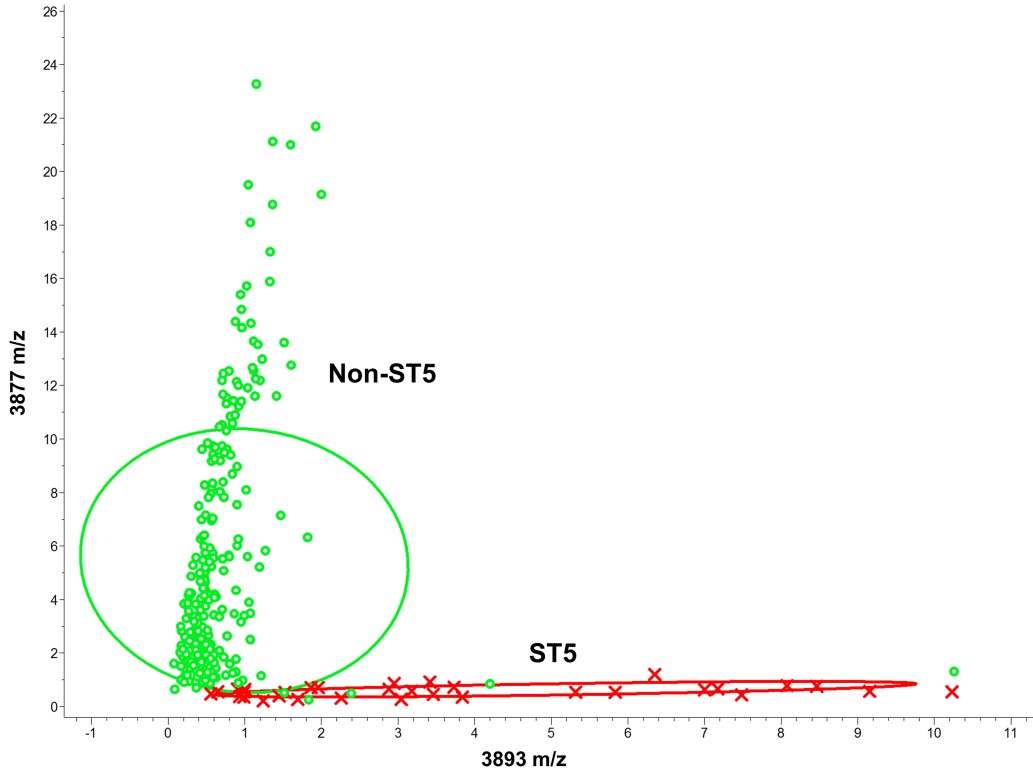

**Figure 3 Scatter plot of the ST5 and non-ST5 isolates.** The peaks at *m/z* 3,893 and *m/z* 3,877 served as the *x*- and *y*-axes, respectively. Intensities of the characteristic peaks were expressed in arbitrary intensity units. The ellipses represent the 95% confidence intervals of peak intensities for ST5 (red ellipse) or non-ST5 (green ellipse). ST5: red crosses, non-ST5: green circles.

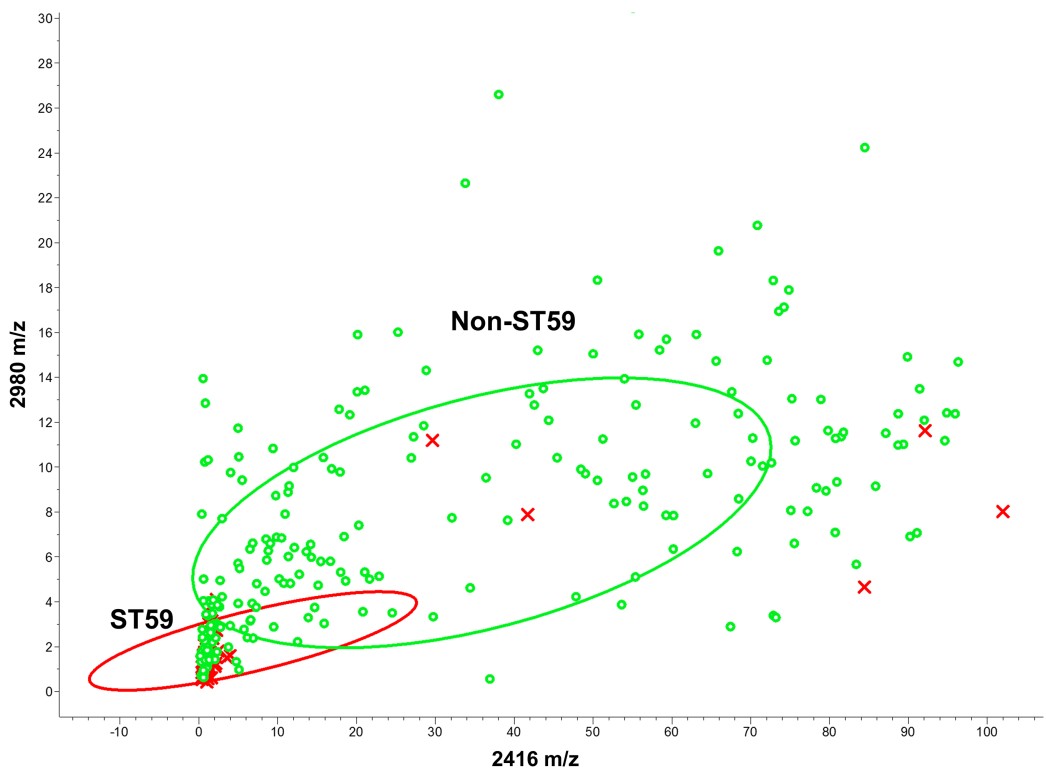

**Figure 4 Scatter plot of the ST59 and non-ST59 isolates.** The peaks at *m/z* 2,416 and *m/z* 2,980 served as the *x*- and *y*-axes, respectively. The intensities of the characteristic peaks were expressed in arbitrary intensity units. The ellipses represent 95% confidence intervals of the peak intensities for ST59 (red ellipse) or non-ST59 (green ellipse). ST59: red crosses, non-ST59: green circles.

## ML models for classification of various MLST types of MRSA

Various MLST types showed specific patterns of peaks expression, as presented in Table 1 and Figs. 3–5. To generate a comprehensive and objective classification, ML algorithms were used. In the ST5 binary classification models, the peaks at *m/z* 3,877 and *m/z* 3,893 showed satisfactory discriminative power (Fig. 3). The QC algorithm attained the highest cross-validation values (94.40%, Table 2).

In ST59 binary classification models, peaks at *m/z* 2,980 and *m/z* 2,416 also showed satisfactory discriminative power (Fig. 4). The GA-KNN algorithm attained the highest cross-validation values among all the algorithms. The GA-KNN algorithm showed highest performance when the number of the nearest neighbor was set as 5 (85.00%, Table 2).

For the binary classification of ST239 vs non-ST239, the peaks at *m/z* 3,277 and *m/z* 6,553 showed moderate discriminative power (Fig. 5). The GA-KNN algorithm showed higher performance over other algorithms when the number of the nearest neighbor was set as 7 (82.80%, Table 2).

In this study, multiclass models for classifying an isolate into one of four lineages were designed. Generally, the GA-KNN algorithm outperformed the other algorithms; more specifically, this algorithm was suitable for detecting ST239 (Table 3). The GA-KNN algorithm could successfully detect the ST5, ST59, and ST239 lineages with an accuracy of

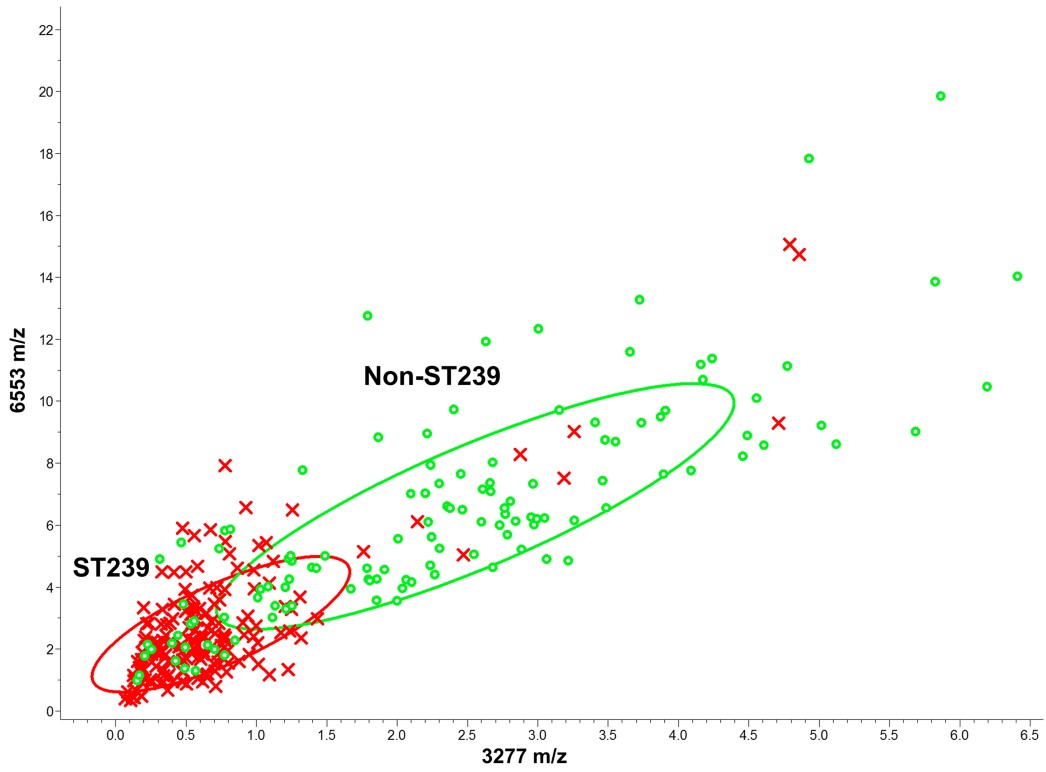

**Figure 5 Scatter plot of the ST239 and non-ST239 isolates.** The peaks at *m/z* 3,277 and *m/z* 6,553 served as the *x*- and *y*-axes, respectively. The intensities of the characteristic peaks were expressed in arbitrary intensity units. The ellipses represent 95% confidence intervals of the peak intensities for ST239 (red ellipse) or non-ST239 (green ellipse). ST239: red crosses, non-ST239: green circles.

>81.00%. However, none of the multiclass models showed reliable accuracy in detecting isolates of the other ST lineages (i.e., ST45, ST241, and ST573).

## DISCUSSION

In this study, the major ST types (i.e., ST5, ST59, and ST239) of blood stream MRSA in Taiwan could be classified through ML-based MALDI-TOF mass spectra analysis with high accuracy. The analysis was conducted using a standardized system (i.e., ClinProTools) to obtain objective and consistent results. Moreover, the ST types of MRSA could be predicted accurately through MALDI-TOF mass spectra analysis before additional molecular typing methods (e.g., MLST). Biomarker peaks of various MRSA ST types were discovered through MALDI-TOF mass spectra analysis by using the ClinProTools software. Some of the biomarker peaks have been previously reported (*Camoez et al., 2016*; *Josten et al., 2013*; *Sauget et al., 2017*; *Zhang et al., 2015*) and validated in the study (namely *m/z* 3,277, 3,877, 3,893, 6,553, and 6,594), whereas some of them have not yet been widely validated (namely *m/z* 2,082, 2,416, 2,430, 2,880, and 2,980).

Nevertheless, reports on the MS characteristics of MRSA are inconsistent (*Lasch et al., 2014*; *Sauget et al., 2017*; *Ueda et al., 2015*). Although the accuracy of typing is sufficient in individual studies, the general patterns of specific lineages are not available

**Table 2  Test performance of various lineage typing ML models.**

| ST | Model | Selected peaks | Acc | Sen | Spe |
|---|---|---|---|---|---|
| ST5 | QC | 8 | 0.94 ± 0.04 | 0.94 ± 0.05 | 0.95 ± 0.05 |
| | SNN | 24 | 0.63 ± 0.05 | 0.91 ± 0.05 | 0.34 ± 0.04 |
| | GA_KNN1 | 30 | 0.83 ± 0.03 | 0.68 ± 0.03 | 0.98 ± 0.05 |
| | GA_KNN3 | 30 | 0.84 ± 0.03 | 0.71 ± 0.03 | 0.97 ± 0.02 |
| | GA_KNN5 | 19 | 0.83 ± 0.03 | 0.68 ± 0.03 | 0.98 ± 0.02 |
| | GA_KNN7 | 14 | 0.88 ± 0.03 | 0.79 ± 0.03 | 0.97 ± 0.02 |
| ST59 | QC | 22 | 0.76 ± 0.03 | 0.85 ± 0.04 | 0.67 ± 0.04 |
| | SNN | 5 | 0.68 ± 0.04 | 0.57 ± 0.05 | 0.79 ± 0.04 |
| | GA_KNN1 | 30 | 0.78 ± 0.03 | 0.62 ± 0.04 | 0.94 ± 0.03 |
| | GA_KNN3 | 29 | 0.82 ± 0.02 | 0.70 ± 0.02 | 0.94 ± 0.03 |
| | GA_KNN5 | 29 | 0.85 ± 0.03 | 0.74 ± 0.03 | 0.96 ± 0.02 |
| | GA_KNN7 | 22 | 0.82 ± 0.02 | 0.67 ± 0.02 | 0.97 ± 0.02 |
| ST239 | QC | 6 | 0.81 ± 0.02 | 0.91 ± 0.01 | 0.72 ± 0.02 |
| | SNN | 1 | 0.58 ± 0.03 | 0.72 ± 0.03 | 0.44 ± 0.04 |
| | GA_KNN1 | 30 | 0.80 ± 0.02 | 0.88 ± 0.03 | 0.72 ± 0.02 |
| | GA_KNN3 | 30 | 0.80 ± 0.02 | 0.85 ± 0.02 | 0.74 ± 0.02 |
| | GA_KNN5 | 29 | 0.81 ± 0.02 | 0.89 ± 0.01 | 0.72 ± 0.02 |
| | GA_KNN7 | 28 | 0.83 ± 0.02 | 0.90 ± 0.02 | 0.76 ± 0.02 |

Notes:
QC, QuickClassifier; SNN, supervised neural network; GA, genetic algorithm; KNN, K-Nearest Neighbor.
The number following "GA-KNN" indicates the number of the nearest neighbors used in models.
Selected peaks: number of peaks selected by the models; Acc, accuracy; Sen, sensitivity; Spe, specificity. Performance metrics are expressed as mean ± standard error.

**Table 3  Accuracy of various multiclass models in classifying different ST lineages.**

| Model | Selected peaks | ST5 | ST59 | ST239 | Others |
|---|---|---|---|---|---|
| QC | 24 | 0.65 ± 0.05 | 0.85 ± 0.03 | 0.79 ± 0.03 | 0.09 ± 0.07 |
| SNN | 25 | 0.65 ± 0.05 | 0.34 ± 0.05 | 0.25 ± 0.06 | 0.13 ± 0.09 |
| GA_KNN1 | 30 | 0.58 ± 0.05 | 0.63 ± 0.04 | 0.84 ± 0.02 | 0.09 ± 0.07 |
| GA_KNN3 | 26 | 0.73 ± 0.03 | 0.78 ± 0.03 | 0.90 ± 0.02 | 0.13 ± 0.08 |
| GA_KNN5 | 27 | 0.73 ± 0.03 | 0.78 ± 0.03 | 0.94 ± 0.02 | 0.04 ± 0.07 |
| GA_KNN7 | 22 | 0.92 ± 0.03 | 0.81 ± 0.03 | 0.94 ± 0.02 | 0.04 ± 0.07 |

Notes:
QC, QuickClassifier; SNN, supervised neural network; GA, genetic algorithm; KNN, K-Nearest Neighbor.
The number following "GA-KNN" indicated the number of the nearest neighbor used in models; Selected peaks: number of peaks selected by the models. Accuracies were expressed as mean ± standard error.

(*Sauget et al., 2017*). Currently, bacterial lineages obtained from geographically diverse areas cannot be clearly discriminated using MALDI-TOF MS. However, more robust characteristic patterns of various types may be available in regions with geographic barriers (e.g., Taiwan Island) than in regions without these barriers. The characteristic patterns of specific lineages may be the result of the disseminated lineages in local areas. By contrast, establishing a localized solution by using an appropriate method of

interpreting mass spectra may be more practical and crucial than establishing a generalized pattern. The isolates in this study were obtained from multiple reference hospitals in Taiwan. Consequently, the isolates used in the study can represent the molecular characteristics of MRSA in the local region. The localized molecular characteristics of MRSA may be sufficiently useful for clinical practice in regions with geographic barriers (e.g., Taiwan Island).

Currently, clinical microbiology laboratories generally use MALDI-TOF MS for bacterial identification because of its advantages in accuracy and effectiveness over traditional biochemical methods (*Lartigue, 2013*). Before analytical measurement by using MALDI-TOF MS, a protein extraction process is necessary. In-tube extraction or the direct deposit method are two common extraction methods used in clinical microbiology laboratories. In-tube extraction provides more purified intracellular components than the direct deposit method does, which results in high quality MALDI-TOF mass spectra and low noise. By contrast, the direct deposition of bacteria onto a steel plate is considered a less labor-intensive and more rapid preanalytical process than in-tube extraction. The time for the entire process and the turnaround time of MALDI-TOF MS can be considerably reduced by using the direct deposit method. Consequently, considering the relevance in routine practice, the direct deposit method was evaluated in this study.

Knowledge of the bacterial molecular type of MRSA is crucial while performing epidemiological studies on bacterial outbreak. In this study, specific peaks were identified for major clonal lineages of MRSA. The peaks *m/z* near 3,277, 3,877, 3,893, 6,553, and 6,594 identified in this study have been reported in previous studies as characteristic peaks in discriminating MRSA clonal complexes (*Camoez et al., 2016*; *Josten et al., 2013*; *Wolters et al., 2011*). *Zhang et al. (2015)* reported the highest expression of the peaks at *m/z* 3,277 and 6,554 in ST59. The peak at *m/z* 6,594 was recognized as the SA1452 protein and as a biomarker of the clonal complex 8 and USA-300 lineages; both these MRSA lineages are prevalent in communities and hospitals (*Boggs, Cazares & Drake, 2012*; *Josten et al., 2013*; *Wolters et al., 2011*). In this study, the peak at *m/z* 6,594 was characteristic for ST239, which were the most prominent ST type of bloodstream MRSA and prominent HA MRSA (Table 1). By contrast, some of the characteristic peaks have not previously been reported and validated as discriminative peaks (i.e., peaks at *m/z* 2,082, 2,416, 2,430, 2,880, and 2,980) (*Sauget et al., 2017*); however, they play a crucial role in MLST type classification. For example, the peak at *m/z* 2,980 was a distinguishing peak for the ST59 MRSA lineages (Fig. 4), which is one of the characteristic ST types of the CA MRSA in Taiwan (*Huang & Chen, 2011*). Moreover, the peaks at *m/z* 2,416 and 2,430 were noted as characteristic peaks in the ST5 and ST239. In the QC models, the peak at *m/z* 2,416 was proven to represent *psm-mec*, which is strongly associated with SCC*mec* III and VIII (*Queck et al., 2009*). Briefly, although the peak at *m/z* 2,416 is commonly expressed peak in MRSA, the expression level, in addition to its presence, may serve as an informative feature in classification of MRSA lineages.

Furthermore, for identifying subtle differences in the MALDI-TOF mass spectra for preliminary reporting of AST or subspecies results, not only single characteristic peaks but also specific combinations of characteristic peaks may be beneficial. Single MS

peaks have provided some characteristics of each major ST type (Table 1; Figs. 1–5). Additional integration of these characteristic peaks by using ML models may generate a more comprehensive and robust result for ST typing of MRSA (Table 2) than that generated using single peaks. Consequently, the accuracy of the subspecies classification by using a combination of peaks may be higher and more resistant to variations in analysis than that using a single peak because of comprehensive interpretation. Zhang et al. (2015) reported the successful use of ML models (by ClinProTools) in analyzing MALDI-TOF mass spectra to classify various MLST types of MRSA. High performance of the binary models (including ST5, ST45, ST59, and ST239 binary classification models) were described. However, the high performance might have resulted from over fitting because only one specific combination of training sets and validation sets was used for performance evaluation.

This study had several limitations. Firstly, our study did not adopt nucA polymerase chain reaction or other sequencing methods to exclude *S. argenteus*, which were previously identified as *S. aureus* using molecular typing (Thaipadungpanit et al., 2015; Tong et al., 2015). The MRSA isolates were collected from an island with geographic barriers, thus resulting in a relative simple ST distribution. Possibly, our method might be unsuitable in other country with wide variety of bacterial lineage. We thus recommend researchers who intend to adopt our approach to train and validate their own model by using regional MRSA lineages. Another limitation of this study is that the MRSA isolates in this study are all from previously collected samples under the SMART program. This indicates the model may not exhibit sufficient accuracy in performance on new MRSA isolated in the future. Moreover, only six MLST types (namely ST5, ST45, ST59, ST239, ST241, and ST573) of MRSA were included for models training and validation. Based on the study design, ML learning models can be used for detecting the major lineages of MRSA in Taiwan only, but they cannot be used for detecting new clones, which were not included in model training. Consequently, the models are designed to be used for reporting preliminary lineage information in outbreak investigation. When a new clone emerges in the future, the ML models could be tuned and updated using the newly collected datasets. Our study has demonstrated that in regions with limited medical resources, an MALDI-TOF-based MLST typing model may serve as a valuable method for timely intervention in the transmission or outbreak of MRSA.

## ACKNOWLEDGEMENT

This manuscript was edited by Wallace Academic Editing.

### Funding

This work was supported by grants from Chang Gung Memorial Hospital (CMRPG3F1081, CMRPG3D1382, CORPG3H0451, CORPG3H0421, CORPG3H0431 and CORPG3H0441) and the Ministry of Science and Technology, Taiwan (MOST-104-2320-B-182A-005-MY3 and MOST-105-2811-B-182A-004). The funders had no role in

study design, data collection and analysis, decision to publish, or preparation of the manuscript.

## Grant Disclosure
The following grant information was disclosed by the authors:
Chang Gung Memorial Hospital: (CMRPG3F1081, CMRPG3D1382, CORPG3H0451, CORPG3H0421, CORPG3H0431 and CORPG3H0441).
Ministry of Science and Technology, Taiwan: MOST-104-2320-B-182A-005-MY3 and MOST-105-2811-B-182A-004.

## Competing Interests
The authors declare that they have no competing interests.

## Author Contributions
- Hsin-Yao Wang conceived and designed the experiments, performed the experiments, analyzed the data, prepared figures and/or tables, authored or reviewed drafts of the paper.
- Frank Lien conceived and designed the experiments, analyzed the data, prepared figures and/or tables.
- Tsui-Ping Liu performed the experiments.
- Chun-Hsien Chen performed the experiments, analyzed the data, prepared figures and/or tables.
- Chao-Jung Chen conceived and designed the experiments, contributed reagents/materials/analysis tools, authored or reviewed drafts of the paper, approved the final draft.
- Jang-Jih Lu conceived and designed the experiments, contributed reagents/materials/ analysis tools, authored or reviewed drafts of the paper, approved the final draft.

## Data Availability
The raw data are provided in the Supplemental File.

## Supplemental Information
Supplemental information for this article can be found online at http://dx.doi.org/10.7717/peerj.5784#supplemental-information.

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
