# Peer review of "Application of a MALDI-TOF analysis platform (ClinProTools) for rapid and preliminary report of MRSA sequence types in Taiwan"

_PeerJ, doi:10.7717/peerj.5784_

## Round 0.1 · original submission · Major Revisions

I have taken into account the reviewer comments. I highlight the request for a direct comparison of MLST compared to MALDI, and that you address Reviewer 1 concerns under the ‘Validity of findings’ section. Similarly, reviewer 2 has requested development of a single model that classifies an isolate into one of the four categories. I agree that this would be more useful than the binary classification and that you should compare this then to MLST.

I also agree with both reviewers that the manuscript needs to be read and corrected by a native English speaker as there are a number of grammatical errors throughout.

Some minor points:
Line 59-60 – re-word
Line 62, 67-68, 79-81, 97-98
Explain further the Anderson-Darling test
What do you mean exactly by geographically ‘barred’?
Lines 216-243 – suggest you summarise these data, and shorten the paragraph substantially
Lines 256-261 – this would be better in the methods rather than the last paragraph of the conclusions.

·

Basic reporting

Line 47: Tone down the statement that the incidence of MRSA is rapidly increasing. At least fort he European countries, the incidence of MRSA infections are stagnating (see Germany, for instance: https://www.eurosurveillance.org/content/10.2807/1560-7917.ES.2015.20.46.30067).

Line 49: Please clarify what you mean by saying „relationship between the results of genotyping from multilocus sequence typing (MLST) and characteristics of MRSA strains“. Do you mean an association between genotypes and certain virulence factors?

Line 50: Use either „strain“ or „lineages“.

Line 53: It is true what is written here but this was the status almost 10 years ago. Nowadays, whole genome sequencing can be done within 1-2 days for less than 100 €. Sequence based typing methods are by far more discriminatory than anything else. So what is the clinical relevance of this study? MALDI-Tof based typing might have some value if transmission can be ruled out. But to confirm transmission, you definitely need more discriminatory power. Why is spa-tying not mentioned here?

The language is not clear and sometimes of poor quality in some sections. For instance Line 61: In what sense were the results varying? Line 62: If you start with „first“, one expects „second“, „third“ and so on. Line 67: What do you mean by „complicated data“? Line 81: write „to“ instead of „for“. Line 87: Do not start a sentence with digits. I strongly recommend that a native English speaker check the text again.

Line 62: Correct to „it may be due to several reasons“

Line 63: Add...“and the discriminatory power of MALDI-TOF might therefore differ when compared between different MRSA lineages.

Lines 137-138: Mention the ST and add (n= ) in brackets right after the ST. The numbers are very unclear, as it is written right now.

Lines 150-155: Add the OR, 95% CI and p-values in brackets for the association of peaks with the respective STs.

Line 158: Why are these data “preliminary”?

Line 161: Distribution of what?

Figures are ok, raw data are supplied.

Experimental design

What are thin inclusion and exclusion criteria fort he MRSA isolates? Indicate the year of collection.

Line 84: How did you rule out that Staphylococcus argenteus was not misidentified as S. aureus. S. argenteus is now considered to be widespread in Asia so this point need to be addressed.

Line 92: Bruker is based in Bremen, not Leipzig.

Lines 103-107: I gain the impression that the authors did not go carefully through the manuscript before submission. In this paragraph, there are so many mistakes that could be easily avoided: aroE not areE, glpF not glp, arcC not araC. Please adhere to the nomenclature for MLST as suggested by Enright et al.

Line 114 (reference): GmbH is probably not the author!

Statistical analysis: please include a section in Materials and methods as well in the results to calculate the performance of MALDI compared to MLST (sensitivity, specificity, PPV, NPV etc.) In the end, the reader wants to know: how likely is the Maldi-typing result true.
At the end of the discussion: add a section on the limitations of the study

Validity of the findings

Line 182-183: I do not agree with the conclusion that MALDI can differentiate the different MLST ST with high accuracy. Even if the p-values suggest a statistical significance, a quick look to figures 3 to 5 clearly show that there is a huge overlap of the spectra (as I would have expected). The only value of MALDI-typing would be to quickly rule out transmission, which cannot be done considering the significant overlap of spectra.

Additional comments

The authors test the performance of MALDI to type major MRSA lineages from Taiwan and compared results from MALDI with ST. Although the study objectives are fine, the authors fail to highlight the clinical impact of this method. In particularly, I do not agree with the conclusions.

·

Basic reporting

Some improvements are needed.

1. The Figure labels for the scatter plots do not seem to match the text in the results section. In the text Figures 2A, 2B, and 2C are mentioned, but the figures are labeled Figure 1, 2, 3, 4, and 5.
2. Can a P value be reported for scatter plots shown in Figures 2 – 5?
3. There are some minor grammatical errors throughout the manuscript- eg line 81 should say “for confirmation” instead of “for confirm”.

Experimental design

Some suggestions are listed below.

1. It appears that the 10 peaks included in Table 1 are present in all strains, and peak intensity, rather than presence/absence was used to differentiate the different major STs. Were there any peaks that were present in only some STs? Peak presence/absence may be a more robust way to differentiate between STs than peak intensity.

2. In Table 2, the authors present the performance of three binary classification models for ST5, ST59, and ST239. Would it be possible to develop a single model that could classify an isolate into one of four categories (ST5, ST59, ST239, or other)? This would be more useful than multiple binary models.

3. It would be helpful to provide a little more context for the statistical tests included- what exactly are you using them to measure? For example, Wilcoxon/Kruskall-Wallis is assessing peak intensity among ST5, ST59, ST239, or other ST. Why was PTTA (Table 1) conducted? You can also state that Anderson Darling test was used to assess data distribution.

Validity of the findings

One minor suggestion to add to the discussion- What would be the ability of this approach to detect the emergence of a new MRSA clone in Taiwan?

Additional comments

No other comments.

---

## Round 0.2 · Minor Revisions

Thank you for the revisions. I agree with the reviewers that our comments have been adequately addressed. Although I am asking for a minor revision, I am basically happy to accept the manuscript, assuming the minor changes are made. The manuscript will not need to go back to the reviewers. Specifically:
Line 90 – I note that the isolate selection is now 15 years old. Can you provide evidence (include in the discussion) that the dominant MLSTs in 2003 are still the dominant MLSTs more recently? At least for ST5, 59 and 239. And this should still be mentioned as a limitation because it is possible that the ST239 strains in 2018 may be different to those from 2003.
And also note reviewer 1 comments - I understand that these are all blood culture isolates. Is any further information available re community vs hospital acquisition, where these convenience samples or consecutive samples, were duplicates excluded?
Line 94-96 – note that S. argenteus cannot be differentiated from S. aureus based on 16S (see Tong IJSEM 2015. I think you will just need to mention a limitation that you were not able to differentiate and exclude S. argenteus (unless you have also done nuc or NRPS PCR / sequencing.
Table 2 - I agree with reviewer 1, it would be better as a single table.
Table 3 – please indicate in the table caption what the performance metric measure is.

·

Basic reporting

I am happy to re-evaluate the changes and corrections made by the authors. Almost all concerns were addressed. In particular, the authors were able to convincingly highlight the significance of MALDI typing compared to WGS in settings with constrain resources. However, although I am not a native English speaker, I still have the impression that some statements could be improved.

Line 74: “Finally, the data obtained from MALDI-TOF mass spectra are relatively complicated and may be analyzed using various methods”. It is still unclear what it means. Does it mean: “Finally, the data obtained from MALDI-TOF mass spectra are complex and need to be analyzed by bioinformatics tools”?

Experimental design

The inclusion criteria are still not clearly stated. If this is a convenience sampling, it should be stated. At least some basic criteria should be mentioned. Inclusion criteria: community or hospital acquired or both? Colonization or infection or both? Exclusion criteria: duplicate isolates or anything else that applies. The authors might mention the criteria as published by Wang et al. 2012 (if applicable).

Line 94-96: 16sRNA gene sequencing is inappropriate to differentiate S. aureus and S. argenteus and the sequences are identical (doi: 10.1099/ijs.0.062752-0). I suggest NRPS or nuc gene PCR (Zhang et al. etc.).

Table 2 seem to contain 3 sub-tables. They can be combined to make it easier to readby just adding one collum and changing the title as follows: test performance of various lineage typing ML models:

ST Model Selected peaks Acc Sen Spe
ST5 QC 8
SNN 24
GA_KNN1
GA_KNN3
GA_KNN5
GA_KNN7
ST95 QC
SNN
GA_KNN1
GA_KNN3
GA_KNN5
GA_KNN7
STXY

Validity of the findings

NA

Additional comments

NA

·

Basic reporting

no comment

Experimental design

no comment

Validity of the findings

no comment

Additional comments

The majority of the comments and questions raised by reviewers have been responded to in a satisfactory manner. Two minor comments/suggestions are provided below:
1. As suggested by reviewer #1, it may be helpful to include some mention of spa typing in the introduction or discussion, as this is often used for characterization of S. aureus.
2. The section on the AD test (Lines 155 - 165) could be shortened. It is enough to say that it is used by ClinProTools to assess for normal distribution of peaks.

---

## Round 0.3 · accepted · Accept

Thanks for your work on this. I am happy to now accept the manuscript.

#